# High-Frequency Audiometry in Women with and without Exposure to Workplace Noise

**DOI:** 10.3390/ijerph18126463

**Published:** 2021-06-15

**Authors:** Eva Mrázková, Martina Kovalová, Zdeněk Čada, Nikol Gottfriedová, Tomáš Rychlý, Michaela Škerková

**Affiliations:** 1Department of Epidemiology and Public Health, Faculty of Medicine, University of Ostrava, 703 00 Ostrava, Czech Republic; martina.kovalova@osu.cz (M.K.); Z19142@student.osu.cz (N.G.); Z15511@student.osu.cz (T.R.); michaela.skerkova@osu.cz (M.Š.); 2Center for Hearing and Balance Disorders, 708 00 Ostrava, Czech Republic; 3Department of Otorhinolaryngology and Head and Neck Surgery, Regional Hospital, 736 01 Havířov, Czech Republic; 4Department of Otorhinolaryngology and Head and Neck Surgery, 1st Faculty of Medicine Charles, University in Prague and Motol University Hospital, Postgraduate Medical School, 121 08 Prague, Czech Republic; zdenek.cada@fnmotol.cz

**Keywords:** audiogram, high-frequency audiometry, hearing loss, noise exposure, workplace noise, conventional pure tone audiometry, noise-induced hearing loss

## Abstract

For this study, high-frequency audiometry was used to compare the hearing thresholds, with respect to age, among women exposed to noise in their working environment, as well as those not exposed to such noise. The cohort comprised 243 women (average age 36.2 years), of which 88 women were employed in a noisy (L_Aeq,8h_ 85–105 dB) workplace, while 155 women did not experience noise. Age categories were determined according to the World Health Organization (Geneva, Switzerland). Hearing thresholds were measured at frequencies of 0.125–16 kHz. Higher hearing thresholds were found in the youngest age groups (18–29 and 30–44 years) among those exposed to noise, as compared to those who were not. The difference in hearing thresholds between the exposed and unexposed groups increased with age, as well as with the frequencies. The highest difference in hearing thresholds for these age categories was measured at 11.25 kHz. The oldest age group (45–63 years) exposed to noise showed lower hearing thresholds than the unexposed group at all frequencies from 4 kHz to 16 kHz. High-frequency audiometry can be used for the early detection of increased hearing thresholds at high frequencies. High-frequency audiometry could be included in preventive programs, especially for younger people exposed to noise, in order to enable earlier detection of noise-induced hearing loss.

## 1. Introduction

At present, conventional pure-tone audiometry is one of the most widely used examination methods for diagnosing hearing defects. In pure-tone audiometry, the hearing of subjects is tested in an acoustically soundproofed chamber, using an audiometer that generates clear tones over the frequency range of 0.125 kHz to 8 kHz, according to current legislative standards. However, the human auditory range is much wider than the frequency range covered by conventional pure-tone audiometry. The human ear is capable of perceiving tones up to 20 kHz. Therefore, high-frequency audiometry is required to examine the hearing threshold from 8 kHz to 20 kHz [1,2].

High-frequency audiometry has been studied for several decades, but the lack of commercially available equipment and the standardization of calibration recommendations have long limited its use [3,4,5]. At present, high-frequency audiometry is performed according to the currently valid standard: IEC 60645-1 Electroacoustics—Audiometric equipment—Part 1. The section for pure-tone and speech audiometry specifies general requirements for audiometers and ensures that a hearing test carried out considering frequencies between 0.125 kHz and 16 kHz on a human ear, performed with different pure-tone audiometers that comply with this standard, provide the same results substantially [6]. The use of high-frequency tone audiometry, however, is now very similar to conventional pure-tone audiometry. Before using the device, a normal hearing threshold should nevertheless first be determined in a selected group of healthy individuals—that is, those in whom the hearing threshold of 25 dB is not exceeded, as measured using conventional pure-tone audiometry [3].

The presumed age-related hearing loss at frequencies ranging from 0.125 kHz to 8 kHz can be calculated using the valid ISO standard: “Acoustics—Statistical distribution of hearing thresholds as a function of age” [7]. Thus, the data currently in use for the determination of the hearing threshold in the ISO standard for otologically healthy people are based on studies conducted in the 1950s, 1960s, and 1970s [8,9]. The drawback is that, so far, no hearing threshold standards have been set for frequencies between 8 kHz and 20 kHz. These values vary greatly, depending on the age of the individual, the type of audiometer used, and the headphones that are used.

Historically, more men have worked in noisy work environments than women. Even today, far more men than women work in places with high-noise risk; however, the proportion of women in such occupations is on the rise. This is mainly due to changes in production technology and, more generally, in the labour sector. Men tend to use hearing protective devices (HPD) much longer, such that the use of HPD is often automatic for them; to the contrary, women often do not use HPD, although they generally take more care in terms of their own health. Some studies have even stated that the prevalence of HPD use in women is up to three times lower than in men [10]. The National Institute for Occupational Safety and Health (NIOSH, Washington, DC, USA) has estimated that 22 million workers in the United States are exposed to hazardous noise levels in their workplaces, while more than 30 million workers in the United States are exposed to chemicals that are ototoxic in humans and thus may cause damage the auditory system. The estimated prevalence of hearing loss among workers exposed to noise is 12% to 25%, depending on the business segment [11].

Current legislation defines a noise level in the working environment as hazardous if a steady or variable noise whose A-weighted equivalent sound pressure level L_Aeq,8h_ reaches or exceeds 85 dB, or an impulse noise whose A-weighted equivalent sound pressure level L_Aeq,8h_ is equal to or greater than 85 dB and whose peak sound pressure level C L_Cpeak_ reaches or exceeds 140 dB [12]. There is a large number of threshold sets, following varying assessments of noise in the workplace and in the urban environment, depending on the length of exposure, properties of the noise, and so on. In the Czech Republic, this issue has been comprehensively covered by governmental regulations on the protection of health against the adverse effects of noise and vibration. The NIOSH recommended exposure level (REL) for noise exposure in the workplace is 85 dB, the same as in the Czech Republic (L_Aeq,8h_ 85 dB). Exposures at or above this value are considered hazardous [11].

Noise-induced hearing loss (NIHL) is a form of sensorineural hearing loss (SNHL), which initially manifests itself as a loss at 4 kHz. As NIHL progresses, the damage spreads to cover all high frequencies. This is tested for by extending the range of frequencies for tone audiometry measurements to values higher than 8 kHz, mostly between 14 kHz and 16 kHz. Hearing loss is usually bilateral and symmetrical; however, when the exposure is acute, damage to only one ear may also be found. There is also a frequent association between tinnitus and otalgia following noise exposure [13].

The aim of this study is to compare the hearing threshold as a function of age in respondents exposed or unexposed to workplace noise at frequencies from 0.125 kHz to 16 kHz using high-frequency audiometry.

## 2. Materials and Methods

The data were collected between 2018 and 2020 in a specialized clinic. The company where the study cohort was employed gave its consent to conduct the study, provided that all applicable ethical standards would be complied with. All respondents completed and signed an informed consent form to be included in the study.

The cohort comprised respondents employed in a noisy workplace, where the A-weighted equivalent sound pressure level L_Aeq,T_ ranged from 87.0 dB to 90.2 dB. The average A-weighted equivalent sound pressure level L_Aeq,T_, measured by a specialized laboratory throughout the 8-hour shift (430 min of working time), was 88.7 dB and the measurement uncertainty was 1.6 dB. As it was higher than the hygienic limit of A-weighted equivalent sound pressure level L_Aeq,T_ (85 dB), the workplace was considered hazardous in terms of noise. The maximum of peak sound pressure level C L_Cpeak_ measured was 110.4 dB. The noise in the workplace was variable. The maximum working time was 2 h, and there was a break after every 2 h, comprising a total of 50 min of breaks per shift. The analysed company produced various types of filters, transmissions, and other components for the automotive industry, and mainly employed women for manual work. The workplace with exposure to noise was an air-conditioned hall, while the filters were manufactured in a separate room. The room had dimensions of 13.2 × 5.9 × 3.2 m, the floor was made of concrete, the ceiling was coffered, and the walls were lined with tiles. Respondents exposed to a noisy working environment were recruited for this study between 2018 and 2020, and their hearing threshold was evaluated. The employer had a long experience with protecting workers from hazards in their working environment. Occupational hygiene inspections were carried out regularly, and the only risk factor in the workplace was noise. Hearing protection was meticulously observed and strictly enforced, in accordance with Czech legislation and the European standard (BS EN352-2:2000—Hearing protectors. General requirements. Level-dependent earmuffs) [14]. This European standard was designed to guide employers, supervisors, and safety advisers. Hearing protectors are personal protective equipment (PPE) designed to reduce the harmful effects of sound and noise on hearing. The hearing protectors were used at the workplace were uncorded polyurethane foam earplugs with a noise reduction rating (NRR) of 33 dB.

The relative number of women and men employed in the noisy workplace depended on the designation and the company workload. The control group consisted of respondents aged 18–63 years who worked in a job without noise exposure (A-weighted equivalent sound pressure level L_Aeq,8h_ < 80 dB), selected randomly from the database of registrations in the outpatient clinic. The control group was the general population, with exposure to noise in public areas, but without exposure to workplace noise. This group was chosen to fulfil the condition requiring the comparison of average hearing thresholds in respondents exposed and unexposed to workplace noise in individual age categories. The history of noise exposure in public areas and leisure activities (for both groups; that is, those with and without noise exposure in the workplace) was not investigated in this work. Both groups lived in the same area and came from similar social groups. Exposure to workplace noise in the current and former workplace was monitored in the respondents. To allow for comparison of our results with those from other surveys, the respondents were divided into age categories according to the World Health Organization (WHO) [15].

The exclusion criteria were disagreement with inclusion in the study and age outside the range of 18–63 years. Another exclusion criterion was male gender, as the designation in the risk category was associated with female employees. The respondents with differences between right and left ears at frequencies from 0.125 kHz to 4 kHz higher than 10 dB were excluded from the study.

Before beginning the examination, the participants were briefed about the process and the principle of the audiometric measurement. Data collection began with taking a brief case history and other data needed for further processing. The respondents with exposition to a noisy working environment were not exposed to noise on the day and the day before the examination. The condition for the respondents before the audiometric examination itself was 24 h of work rest and no noise exposure. This was followed by tympanometry (Madsen Zodiac Diagnostic, type 1096), conventional pure-tone audiometry, and high-frequency audiometry (Madsen Astera 2, Headset Sennheiser HDA300), performed according to the standards EN ISO 8253-1:2010 Acoustics—Audiometric test methods and EN ISO 266:1997 Acoustics—Preferred frequencies [16,17]. These standards were last reviewed and confirmed in 2018 and 2021; as such, these versions can be considered current. All instruments were calibrated before the beginning of the measurement. The examination was performed in an acoustic chamber and was administered by the same personnel.

The respondents were fitted with headphones, through which tones of different intensities and frequencies were played, first into one ear, then into the other ear. The measurement started with the ear that the respondent identified as the one with better hearing. If the respondent did not perceive a subjective difference of hearing between the ears, the measurement began in the left ear. The hearing threshold was measured at 0.125, 0.25, 0.5, 0.75, 1, 1.5, 2, 3, 4, 6, 8, 9, 10, 11.25, 12.5, 14, and 16 kHz. The result was plotted as an audiogram, where the hearing loss was expressed in dB (decibel hearing level dB HL) for each frequency and separately for each ear. The average values of hearing loss for individual frequencies were calculated and the average threshold curves for individual age categories were compiled. They were statistically compared in order to determine the difference in hearing threshold (in dB), as well as its statistical significance.

### Statistical Methods

The results recorded in the audiogram were exported to Microsoft Office Excel 2017 (MS Excel; Microsoft Corporation, Washington, DC, USA) for calculation of statistics, following which tables and graphs were drawn. Statistical comparison of data was performed using basic descriptive statistics, the chi-square test, the Student’s *t*-test, the Wilcoxon paired test, and the Mann–Whitney test. Statistical significance was analyzed using the Stata version 13 software (Data Analysis and Statistical Software; StataCorp LP, College Station, TX, USA) and the online program Openepi version 3.01 (Open Source Epidemiologic Statistics for Public Health; Epi Info Development Team, Atlanta, GA, USA). The significance level for testing was set at 5%.

## 3. Study Cohort

A total of 88 respondents exposed to a noisy working environment were recruited over the period from 2018 to 2020. The exposed group comprised a total of 88 women employed in the noisy working environment, while the control group included 155 women without exposure to workplace noise. A total of 243 women were examined. The average age in the whole group was 36.2 years (SD 11.79; min. 18; max. 63). Respondents not exposed to workplace noise were more than 5 years younger, on average (34.3 years; SD 11.89; min. 18; max. 63), compared to those in the exposed group (39.7 years; SD 10.79; min. 19, max 61). This age difference between the exposed and unexposed groups of women was statistically significant (*p* < 0.001). The group was then divided into WHO age categories: 18–29 years, 30–44 years, 45–59 years, and 60–74 years. The age category of 60–74 years was merged with the 45–59 category, as the number in this category was too low; thus, the oldest age category was 45–63 years. This category (45–63 years) was the least represented, with a total of 61 women. The second most represented age category was 30–44 years with 87 women, while the most numerous category was the youngest (18–29 years), with a total of 95 individuals (the sizes of groups with and without noise exposure in the workplace are given in Table 1). A statistically significant difference, with regard to exposure or non-exposure to workplace noise, was found between the distribution of women in individual categories (*p* = 0.0004). The average working time at risk of noise was 5.69 years (SD 4.50; min. 0.5; max 26). The youngest age category (18–29 years) worked in a noisy workplace for an average of 3.15 years (SD 1.50; min. 1; max. 6). The second age category (30–44 years) worked in a noisy workplace for an average of 5.08 years (SD 3.50; min. 0.5; max. 15.5), while the oldest age category (45–63 years) worked in a noisy workplace for an average of 8.26 years (SD 5.65; min. 2; max. 26). The group was then divided, for further analysis, into categories according to noise exposure: exposure less than 5 years (average age in this group was 36.67 years, SD 10.79; min. 19; max. 60), exposure 5–10 years (average age in this group was 40.48 years, SD 9.43; min. 25; max. 60), and exposure more than 10 years (average age in this group was 48.43 years, SD 7.68; min. 37; max. 61).

## 4. Results

The average threshold curves for individual age categories are shown in Figure 1. It is obvious that, with increasing age, there is a gradual deterioration of the hearing threshold, regardless of exposure to workplace noise. Nevertheless, hearing thresholds were more pronounced in the exposed groups (except for the age category 45–63 years). These differences were most pronounced in the higher measured frequencies (8–12.5 kHz). The results of comparison of the average values of respondents exposed and unexposed to workplace noise, regardless of age, showed that the hearing thresholds were higher at all frequencies for exposed groups aged 18–29 years and 30–44 years. Table 1 and Figure 1 show the variation of hearing threshold values for the individual frequencies in the exposed and unexposed groups.

In the 18–29 years age group, the biggest differences were observed at the frequencies 11.25 kHz and 12.5 kHz. The hearing threshold in exposed persons was higher for almost all measured frequencies (except for 4 kHz). Nevertheless, the average losses at frequencies of 0.125–16 kHz did not exceed 20 dB in both the exposed and unexposed groups. For exposed respondents, the highest average loss was 20 dB at 16 kHz. The difference in the calculated average values of the hearing threshold in this age category ranged from 0 dB to 6 dB. The largest difference (6 dB) was at 11.25 kHz, where the average hearing loss was calculated to be 11 dB for the exposed group and 5 dB for the unexposed group. A statistically significant difference was found between the average hearing thresholds at 0.75 kHz, 6 kHz, 11.25 kHz, and 12.5 k Hz frequencies for average hearing losses in dB. No statistically significant differences were found for the other frequencies. At the frequencies of 0.5 kHz and 9 kHz, no statistically significant differences were found, but the calculated *p*-value approached 0.05. The calculated *p*-values for average hearing losses for the individual frequencies are given in Table 2.

In the 30–44 years age group, the average hearing losses did not exceed 20 dB in either the exposed or the unexposed group at any measured frequency up to 11.25 kHz. At higher frequencies—above 12.5 kHz—the average threshold curves gradually sloped upward, with higher values being reached for the exposed group. The highest differences (4 dB) between the exposed and unexposed groups were at the frequencies 9–12.5 kHz. Statistically significant differences were found at 2 kHz, 3 kHz, 8 kHz, 9 kHz, 11.25 kHz, and 12.5 kHz. No statistically significant difference at 6 kHz was found, but the calculated value approached 0.05 (Table 2).

In the oldest age group (45 to 63 years), the differences in average hearing losses were the highest, ranging from 0 dB to 7 dB. As in the other age groups, the highest differences in average hearing losses between the individual groups were at 11.25 kHz and 14 kHz. The hearing loss at 11.25 kHz was 23 dB for exposed respondents and 30 dB for unexposed respondents, such that the difference in these values was as large as 7 dB. The same difference of average hearing loss was observed at 14 kHz: 50 dB for exposed respondents and 57 dB for unexposed respondents. Statistically significant differences were found between the average hearing loss of exposed and unexposed respondents at 1.5 kHz (*p* = 0.042). Average values, standard deviations, and minimum and maximum hearing losses for the individual frequencies in the exposed and unexposed groups are given in Table 1.

The average hearing losses in unexposed persons did not exceed 20 dB in either the exposed or the unexposed group at any measured frequency up to 12.5 kHz. The difference in average hearing loss between unexposed and exposed individuals also increased with an increase in the frequency of the test sound. The highest differences in average hearing thresholds between the unexposed groups and all groups with exposure to noise were at the frequencies of 14 kHz and 16 kHz. The hearing threshold at 14 kHz was 31 dB for respondents exposed <5 years, 35 dB for respondents exposed to noise 5–10 years, 49 dB for respondents exposed >10 years, and 28 dB for unexposed respondents; therefore, the difference in these values was as large as 21 dB. Hearing thresholds at 16 kHz were 48 dB for <5 years exposed respondents, 41 dB for 5–10 years exposed respondents, 51 dB for >10 years exposed respondents, and 33 dB for unexposed respondents. The highest difference was in the group with the longest exposure: 18 dB. Average values, standard deviations, and minimum and maximum hearing losses for the individual frequencies in the exposed groups (depending on the duration of exposure) and unexposed group are given in Table 3. Statistically significant differences were not found between the average hearing thresholds of those exposed <5 years and unexposed respondents only at the frequencies 0.125 kHz, 0.25 kHz, 3 kHz, 4 kHz, 14 kHz, and 16 kHz. At the frequency of 1 kHz, no statistically significant difference was found, but the calculated *p*-value approached 0.05. Statistically significant differences were found between the average hearing thresholds of those exposed for 5–10 years and unexposed respondents at the frequencies of 0.75 kHz, 3 kHz, 6 kHz, 8 kHz, 9 kHz, 11.25 kHz, and 12.5 kHz. At the frequency of 10 kHz, no statistically significant difference was found, but the calculated *p*-value approached 0.05. In the group with the longest exposure to noise, statistically significant differences were found at the frequencies 0.75–3 kHz and 9–12.5 kHz. No statistically significant difference was found at 6 kHz, but the calculated *p*-value approached 0.05 (Table 4). Figure 2 shows the variance of hearing threshold values for the individual frequencies in the exposed groups (depending on the duration of exposure) and the unexposed group.

## 5. Discussion

High-frequency audiometry is used mainly for the early detection of hearing loss. For instance, it is used to test people working in a noisy environment, undergoing treatment with ototoxic drugs, or when examining the hearing threshold in patients with tinnitus, in whom a hearing impairment has not been detected by conventional pure-tone audiometry. It serves mainly for the early detection of hearing impairment, manifested as lowered hearing threshold at high frequencies. The hearing loss becomes first apparent at high frequencies, before it spreads to frequencies essential for communication (i.e., 1–4 kHz). The method for high-frequency audiometric examination does not differ from that used in conventional pure-tone audiometry. For example, in treatments using ototoxic drugs, high frequencies are the first to become damaged. When monitoring hearing threshold damage with ototoxic drugs, the hearing thresholds are measured for each person separately and compared individually with an initial hearing threshold. Therefore, this method can be employed successfully, even though normative hearing thresholds have not been specified [3,18,19].

In our study, the average hearing thresholds from 0.125 kHz to 16 kHz were compared, depending on age and length of exposure to noise at work (number of years at risk of noise at work) for women exposed and unexposed to workplace noise. Statistically significant differences were found between the average hearing thresholds of exposed and unexposed respondents at the frequencies of 0.75 kHz, 6 kHz, 11.25 kHz, and 12.5 kHz in the 18–29 years age group; at 2 kHz, 3 kHz, 8 kHz, 9 kHz, 11.25 kHz, and 12.5 kHz in the 30–44 years age group; and at 1.5 kHz in the 45–63 years age group. No statistically significant difference was found for a few frequencies, but the corresponding calculated *p*-values approached the statistically significant value of 0.05 (i.e., in the youngest group at the frequencies 0.5 kHz and 9 kHz, and in the 30–44 years age group at 6 kHz). Furthermore, in this study, we compared the hearing threshold of unexposed persons with persons exposed to noise at work for less than 5 years, 5–10 years, and more than 10 years. Statistically significant differences were not found between the average hearing threshold of those exposed for less than 5 years and unexposed persons only at the frequencies 0.125 kHz, 0.25 kHz, 3 kHz, 4 kHz, 14 kHz, and 16 kHz. In the group exposed to noise for 5–10 years, statistically significant hearing thresholds were found at the frequencies 0.75 kHz, 3 kHz, 6 kHz, 8 kHz, 9 kHz, 11.25 kHz, and 12.5 kHz; while those for the group exposed to workplace noise for more than 10 years were observed at the frequencies from 0.75 kHz to 3 kHz and 9 kHz to 12.5 kHz. At a few frequencies, statistically significant differences were not found, but the calculated *p*-values approached 0.05 (i.e., in the group exposed less than 5 years at 1 kHz, in the group exposed for 5–10 years at 10 kHz, and in the group with exposure for more than 10 years at 6 kHz).

The standard EN ISO 1999: 2013 provides procedures for estimating the hearing loss due to noise exposure of populations free from auditory impairment, other than that due to noise (with allowance for the effects of age), or of unscreened populations whose hearing capability has been measured or estimated. Persons regularly exposed to noise can develop hearing loss of varying severity. This International Standard can be applied to calculate the risk of sustaining hearing loss due to regular occupational noise exposure, or due to any daily repeated noise exposure. Consequently, this International Standard does not stipulate a specific formula for assessment of the risk of impairment, but specifies uniform methods for the prediction of hearing loss, which can be used for the assessment of impairment, according to the formula desired or stipulated in a specific country. As noise-induced hearing loss is the result not only of occupational noise exposure but also of the total noise exposure of the population, it may be important to take the non-occupational exposure of individuals (e.g., during commuting to and from their jobs, at home, and during recreational activities) into account. Only if this non-occupational exposure is negligible—compared with the occupational exposure—does this International Standard allow for prediction of the occurrence of hearing loss due to occupational noise exposure. Otherwise, it should be used to calculate the hearing loss to be expected from the combined (occupational plus non-occupational) total daily noise exposure. The contribution of the occupational noise exposure to the total hearing loss can then be estimated, if desired. All sound pressure levels stated in this International Standard do not consider the effect of hearing protectors, which would reduce the effective exposure levels and modify the spectrum at the ear. In our study, both employers and workers followed the rules for hearing protection from noise in the workplace. Apart from noise and age, there were no other risks affecting their hearing threshold in the workplace. Possible noise hearing loss was not evaluated, but the current noise hearing loss in the population exposed to work noise and unexposed population living in the same conditions was evaluated, with respect to high-frequency audiometry [20].

Kovalová et al. monitored and compared the average hearing thresholds of 2698 women exposed and unexposed to workplace noise, of whom 479 were exposed to noise in the work environment and the remaining 2219 women formed a control group without noise exposure. They found no statistically significant differences between the average hearing loss, at all examined frequencies, for women under the age of 44. From the graphs presented in their study, it is clear that the average hearing loss in the 18–29 years and 30–44 years age groups for exposed and unexposed subjects did not exceed 20 dB; thus, they were normograms in both monitored groups [21]. These results were confirmed in the present study. Using conventional pure-tone audiometry, mean hearing loss in exposed and unexposed women aged 18 to 29 years ranged from 4 to 12 dB and ranged from 7 to 14 dB in those aged 30 to 44 years at frequencies from 0.125 kHz to 8 kHz. In the oldest age group (45–63 years), average hearing losses exceeding 20 dB with conventional pure-tone audiometry were not observed. However, Kovalová et al. reported average hearing losses of up to 30 dB, both for women employed in noisy workplaces and for those not employed in a noisy workplace. This increase in average hearing loss above 20 dB occurred at frequencies of 4 kHz, 6 kHz, and 8 kHz [21].

Silva et al. reported statistically significant differences in hearing loss, as measured by conventional tonal audiometry, between young adults and older adults, even among non-hearing-impaired respondents. The respondents in that study were divided into only two age groups: 25–35 years and 45–55 years [22]. The same results were obtained in our study, which included respondents with normograms from 0.125 kHz to 8 kHz. Additionally, Silva et al. performed high-frequency audiometry measurements and reported differences between the first and second age groups in the hearing threshold at high frequencies (above 8 kHz). These differences were more pronounced at higher frequencies than at the frequencies examined in conventional pure-tone audiometry [22]. These data also agree with the results obtained in this study.

Silva et al. found that, for the 25–35 years age group, the average hearing loss at high frequencies did not exceed 20 dB; however, in the older age group (45–55 years), the average hearing loss reached up to 50 dB. The highest losses were calculated from measurements performed at 12.5 kHz [22]. These values cannot be accurately compared with the results obtained in our study, as the age group divisions do not match. However, the conclusion that the hearing threshold increases with increasing age, and that this increase is larger at high frequencies, is in general agreement with our results.

Maccà et al. compared average hearing losses at frequencies up to 18 kHz, with respect to age and noise exposure. They monitored 113 individuals exposed to noise and 148 people without noise exposure. The results of their study confirmed the dependence of high-frequency hearing thresholds on age, but also found higher high-frequency hearing thresholds in people exposed to noise, compared to the average hearing loss in people without noise exposure. The largest differences in average hearing threshold values between exposed and unexposed groups were at 4 kHz, 6 kHz, and 14 kHz. They also found more pronounced differences between high-frequency hearing thresholds at and above the age of 30. Greater differences in the average hearing loss at high frequencies among the 30–44 and 45–65 years age groups were not found, compared to that in the 18–29 years age group. Differences in the average hearing loss at high frequencies ranged from 1 to 3 dB (in the 18–29 years age group), 0 to 3 dB (in the 30–44 years age group), and 1 to 3 dB (in the 45–65 years age group). Maccà et al. also observed an average working time at risk of noise of 17.62 years (SD 10.85; min. 0.5; max. 47). In our study was observed an average working time at risk of noise 5.69 years (SD 4.50; min. 0.5; max. 26) [23].

A Scottish study of hearing loss in female weavers in 1965, cited by Kovalová et al. [21], also took into account the noise exposure time, which was much higher than in the study of Maccà et al. [23]; however, all studies confirmed a worsening of the hearing threshold, depending on the duration of noise exposure at the workplace [21,22,23].

In this study, a shorter average time of noise exposure was found, but there was an evident trend between time spent exposed to noise risk, increasing age, and increase in the hearing threshold, especially at high frequencies. The threshold of respondents increased with age and exposure to noise in the workplace. All respondents with exposition to workplace noise used HPDs throughout all the time with noise exposure at work. However, the correctness of effective use by the respondents and the wear of HPDs has not been checked by the authors. The wear of HPDs could explain the increase in hearing threshold levels in women with exposition to workplace noise. The employer should check the correct use of the HPDs as rigorously as the noise in the workplace itself.

A multivariate analysis by Van Wieringen et al. showed that age was the primary predictor of change in hearing thresholds in the high frequency range (10–18 kHz), followed by noise exposure. This conclusion was also supported by our results, where the highest differences in the average hearing threshold among exposed respondents in the youngest age group (18–29 years) were found at the frequencies of 11.25 kHz and 12.5 kHz (6 dB). The analysis also showed the opposite for the conventional 0.25–8 kHz frequency range, where noise exposure was the primary predictor. The results of this study suggest that high-frequency audiometry could be used as an early indicator of hearing loss due to noise and acoustic trauma, rather than audiometry at the conventional frequency of 4 kHz, especially for younger groups [24].

The effect of age and noise exposure on the hearing threshold at high frequencies has also been investigated by Ahmed et al. in a cross-sectional study with 187 respondents exposed to noise and 52 respondents without noise exposure at work. The hearing thresholds of all monitored subjects were measured at frequencies ranging from 0.25 kHz to 18 kHz [25]. The results of their study were in agreement with those obtained in our study.

The average hearing loss, in the study by Ahmed et al., increased with age at high frequencies. Exposed subjects had significantly higher hearing thresholds than those not exposed at all the high frequencies tested, where the difference between the two groups was greatest at 14 kHz. Higher hearing thresholds in the unexposed group were found only for the oldest group at frequencies over 4 kHz [25].

Other observed values related to the tendency of the thresholds to deteriorate with increasing frequency in both groups. This finding, presenting a descending configuration of the audiometric curve, can be explained by the fact that the human ear requires more acoustic energy to detect higher frequencies. However, a significant deterioration at 14 and 16 kHz indicates a further effect on cochlear function, which significantly increases the hearing thresholds at these frequencies in the tinnitus group [26]. A similar phenomenon was observed in our study, where the increase in hearing loss depended on age and increasing frequency in the unexposed population, and more significantly in the exposed population (except for the age group of 45–63 years).

Rocha et al. concluded that noise in the work environment interferes with high frequency thresholds. All mean hearing threshold values found in the experimental group exposed to noise were higher than in the control group (i.e., not exposed to noise). They suggested that these data reinforce the importance of high-frequency studies, even in conventional pure-tone audiometry, for the early detection of noise-induced hearing loss. A group of 92 firefighters aged 30–49 years were included in this study. Although in this case they were all men, the results correlated with the results of other studies [27]. Therefore, with increasing age, the hearing threshold increases at high frequencies, where hearing loss at these frequencies is also affected by noise exposure [24,27]. This conclusion was supported by our results. As was found, the highest differences in the average hearing threshold were among exposed women in the 30–44 years age group, but not in those aged 45–63 years. In this latter group, the hearing thresholds above 4 kHz were higher in unexposed persons than in the exposed persons. Presbycusis is multifactorial, and noise is just one factor among many influencing hearing impairment.

In a study by Garcia et al., hearing loss was observed at higher frequencies compared to conventional pure-tone audiometry, with the range from 9 kHz to 11.25 kHz being the most affected. These results indicate that high-frequency audiometry could be useful in the early diagnosis of acoustic trauma in those who have not yet been affected in conventionally investigated frequencies [13]. These conclusions were also confirmed by the results of our work. The largest difference between average hearing thresholds in the groups aged 18–29 and 30–44 were found at frequencies between 9 kHz and 12.5 kHz. However, at the frequencies of 14 kHz and 16 kHz, no differences in hearing thresholds between the exposed and unexposed women were found.

In some cases, high-frequency audiometry can be used to distinguish between hearing loss caused by noise and presbycusis. In presbycusis, the auditory threshold progressively increases with the higher frequency range. With noise-induced hearing loss, the hearing threshold usually increases first in the 4–6 kHz range, while there is a marked improvement at 10, 12.5, or 14 kHz [4].

Sheppard et al. have stated that the prescribed regulations for the protection of hearing from adverse noise in the work environment have not been appropriately designed to protect the entire population working in noise. Rather, they are designed to protect a substantial portion of professional noise-induced hearing loss while maintaining a feasible cost–benefit ratio. There is also a growing awareness in the population that noise exposure, although mild, can contribute to a number of other auditory and non-auditory consequences, which may not be identified in hearing protection monitoring programs [28]. The development of audiological technology and the gradual availability of high-frequency audiometry could protect against hearing loss caused by noise, especially in younger people. Hearing loss in younger people can be detected by high-frequency audiometry before the speech frequencies are affected.

The standardization of normal hearing losses at high frequencies, considering the results of a number of studies published so far, remains controversial. It is, therefore, necessary that studies are performed using the same methodology so better data comparison and standard thresholds can be derived. Given that many of these are variable (e.g., standardization and calibration of the device used, limitations of the measuring equipment, calibration, type and placement of headphones, and age and gender differences), the methodology needs to be the same so that a reliable comparison of studies can be made [27].

## 6. Conclusions

At present, conventional pure-tone audiometry is widely employed to measure hearing thresholds. This method measures hearing loss at frequencies from 0.125 kHz to 8 kHz, even though the human auditory field is much wider—ranging from 0.016 kHz to 20 kHz. High-frequency audiometry is used to measure hearing loss at frequencies from 8 kHz to 20 kHz. As there exist no high-frequency hearing loss standards, high frequency audiometry is not currently used to diagnose hearing impairment, and is mainly used only for the early detection of increased hearing thresholds at high frequencies.

In this work, the average hearing thresholds from 0.125 kHz to 16 kHz were compared between female respondents exposed and unexposed to workplace noise. Hearing loss was higher in exposed respondents compared to the unexposed respondents in the younger age groups (18–44 years). This comparison was made for each age group separately, as the hearing threshold at high frequencies increases with age. This should be taken into account in the future when setting normative values, which should, therefore, be set separately for each age group. The results of this work also confirmed that the hearing threshold increases, depending on the duration of noise exposure.

In the future, it would be desirable to expand the possibilities of using high-frequency audiometry by setting normative standards. In the meantime, when monitoring noise damage, it is necessary to measure the auditory thresholds for each person separately and compare them individually with the initial hearing threshold. Hearing loss due to workplace noise can be prevented relatively simply. The health and safety of employees in the workplace is regulated by legislation, and observance of these regulations, their enforcement, and early diagnosis are necessary and can be highly effective. Especially with the increasing age of workers, early detection of and compensation for any hearing loss is critical. One of the possibilities could be to employ high-frequency audiometry for preventive examinations in people exposed to noise in order to enable the early detection of damage or hearing loss as a result of such noise exposure.

## Figures and Tables

**Figure 1 ijerph-18-06463-f001:**
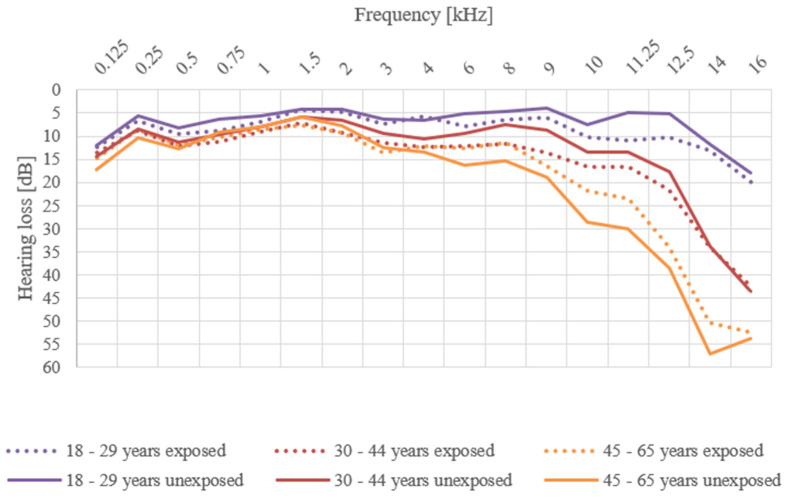
Average hearing loss of women exposed or unexposed to workplace noise.

**Figure 2 ijerph-18-06463-f002:**
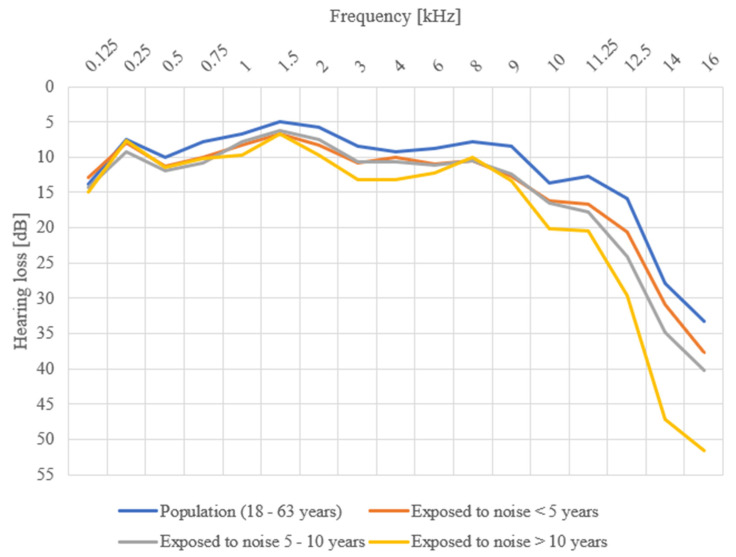
Average hearing thresholds of women exposed and unexposed to workplace noise, depending on the length of work-related exposure to noise.

**Table 1 ijerph-18-06463-t001:** Average values, standard deviations, and minimum and maximum hearing thresholds of women exposed and unexposed to workplace noise.

Age Group (Number)	AHT Right + Left Ears	Frequency (kHz)																
		0.125	0.25	0.5	0.75	1	1.5	2	3	4	6	8	9	10	11.25	12.5	14	16
18–29 y Exposed N = 13	AHT	12	7	10	9	7	4	5	7	6	8	6	6	10	11	10	13	20
	(SD)	(4.13)	(4.39)	(3.56)	(4.9)	(4.99)	(4.36)	(3.62)	(4.48)	(3.73)	(5.61)	(6.56)	(4.95)	(8.46)	(9.85)	(9.44)	(13.06)	(16.1)
	Min.	5	0	5	2.5	0	0	0	0	0	0	0	0	0	0	0	0	0
	Max.	20	15	15	20	17.5	17.5	12.5	15	12.5	25	22.5	20	27.5	27.5	35	47.5	55
30–44 y Exposed (N = 33)	AHT	14	9	12	11	9	7	9	11	12	12	12	13	17	17	22	34	43
	(SD)	(4.51)	(4.83)	(4.94)	(5.03)	(5.52)	(5.79)	(5.5)	(5.31)	(6.61)	(8.47)	(7.99)	(8.28)	(8.99)	(11.22)	(14.57)	(17.83)	(13.73)
	Min.	0	0	5	2.5	0	0	0	5	0	0	2.5	0	5	0	2.5	10	10
	Max.	27.5	27.5	27.5	30	30	30	30	30	30	42.5	45	42.5	47.5	50	60	65	62.5
45–65 y Exposed (N = 38)	AHT	15	9	12	10	8	8	9	13	12	13	11	16	22	23	34	50	52
	(SD)	(4.24)	(3.87)	(5.04)	(4.47)	(4.55)	(4.87)	(4.18)	(6.1)	(6.02	(6.86)	(6.67)	(11.7)	(8.8)	(12.03)	(15.01)	(13.31)	(10.99)
	Min.	7.5	2.5	0	5	0	0	2.5	0	0	0	0	5	7.5	5	10	17.5	17.5
	Max.	20	17.5	20	20	17.5	25	17.5	27.5	27.5	30	27.5	57.5	47.5	55	60	75	62.5
18–29 y Unexposed (N = 10)	AHT	12	6	8	6	6	4	4	6	7	5	5	4	8	5	5	12	18
	(SD)	(3.69)	(3.56)	(4.49)	(4.55)	(4.58)	(4.84)	(5.37)	(4.53)	(5.29	(7.72)	(7.29)	(7.11)	(6.55)	(5.72)	(6.54)	(9.47)	(13.63)
	Min.	0	0	0	0	0	−5	−5	−2.5	−5	−5	−7.5	−10	−2.5	−5	−10	−5	−2.5
	Max.	20	15	20	22.5	25	20	27.5	17.5	30	60	50	42.5	37.5	25	27.5	42.5	55
30–44 y Unexposed (N = 10)	AHT	14	8	11	10	8	6	7	9	11	9	8	9	13	13	18	34	43
	(SD)	(4.56)	(5.69)	(5.23)	(5.95)	(5.43)	(5.99)	(5.54)	(5.18)	(6.42	(7.58)	(7.76)	(9.18)	(9.41)	(13.04)	(18.66)	(20.97)	(13.56)
	Min.	2.5	0	2.5	0	(2.5	0	0	0	0	0	−2.5	−5	0	−2.5	0	0	2.5
	Max.	27.5	30	32.5	35	32.5	27.5	20	22.5	30	37.5	40	42.5	47.5	52.5	70	77.5	62.5
45–65 y Unexposed (N = 10)	AHT	17	10	13	9	8	6	8	12	13	16	15	19	29	30	39	57	54
	(SD)	(5.49)	(5.83)	(4.81)	(5.77)	(4.67)	(5.18)	(5.91)	(6.58)	(7.61	(9.67)	(13)	(17.09)	(19.06)	(21.36)	(21.2	(14.69)	(6.22)
	Min.	7.5	0	2.5	0	0	0	0	2.5	2.5	0	0	0	2.5	2.5	7.5	20	37.5
	Max.	30	25	27.5	30	25	22.5	27.5	27.5	37.5	35	50	62.5	62.5	67.5	77.5	82.5	62.5

AHT, average hearing thresholds (dB); SD, standard deviation.

**Table 2 ijerph-18-06463-t002:** Comparison of the average hearing thresholds of respondents exposed and unexposed to workplace noise (in terms of *p*-values).

Age Group		Frequency (kHz)																
		0.125	0.25	0.5	0.75	1	1.5	2	3	4	6	8	9	10	11.25	12.5	14	16
18–29 y	Average hearing thresholds (dB)	NS	NS	NS (0.063)	0.040	NS	NS	NS	NS	NS	0.013	NS	NS (0.063)	NS	0.029	0.037	NS	NS
30–44 y		NS	NS	NS	NS	NS	NS	0.010	0.038	NS	NS (0.067)	0.010	0.008	NS	0.013	0.009	NS	NS
45–65 y		NS	NS	NS	NS	NS	0.042	NS	NS	NS	NS	NS	NS	NS	NS	NS	NS	NS

NS: not significant—the result of the Mann–Whitney test was >0.05, The level of statistical significance was set at 5%.

**Table 3 ijerph-18-06463-t003:** Average values, standard deviations, and minimum and maximum hearing thresholds of women unexposed and exposed to noise, depending on the length of work-related exposure to noise.

Age Group (Number)	AHT Right + Left Ears	Frequency (kHz)																
		0.125	0.25	0.5	0.75	1	1.5	2	3	4	6	8	9	10	11.25	12.5	14	16
18–65 y Unexposed Women (N = 155)	AHT	14	7	10	8	7	5	6	8	9	9	8	8	14	13	16	28	33
	(SD)	(4.81)	(5.14)	(5.14)	(5.48)	(4.99)	(5.33)	(5.7)	(5.72)	(6.74)	(9.15)	(9.71)	(11.92)	(13.61)	(15.88)	(19.57)	(13.06)	(19.66)
	Min.	0	0	0	0	−2.5	−5	−5	−5	−5	−5	−7.5	−10	−2.5	−5	−10	−5	−2.5
	Max.	30	30	32.5	35	32.5	27.5	27.5	27.5	37.5	60	50	62.5	67.5	67.5	77.5	82.5	62.5
Noise at work <5 years (N = 47)	AHT	13	8	11	10	8	7	8	11	10	11	10	13	16	17	21	31	48
	(SD)	(4.36)	(4.56)	(4.13)	(4.39)	(4.69	(5.21)	(4.88)	(5.76)	(6.28)	(7.93)	(8.17)	(10.66)	(9.71)	(11.47)	(16.05)	(20.51)	(18.36)
	Min.	0	0	5	2.5	0	0	0	0	0	0	O	0	0	0	0	0	2.5
	Max.	20	17.5	20	20	20	25	17.5	25	27.5	42.5	45	57.5	47.5	50	60	65	62.5
Noise at work 5–10 years (N = 28)	AHT	14	9	12	11	8	6	7	10	11	11	10	12	16	17	24	35	41
	(SD)	(4.88)	(4.97)	(5.98)	(5.84)	(6.12	(6.25)	(5.99)	(5.86)	(6.12)	(7.37)	(7.51)	(9.07)	(10.24)	(13.79)	(16.54)	(19.04)	(18.28)
	Min.	5	5	0	2.5	0	0	0	0	0	0	0	5	0	0	0	2.5	0
	Max.	27.5	27.5	27.5	30	30	30	30	30	30	32.5	27.5	42.5	47.5	55	60	65	62.5
Noise at work >10 years (N = 13)	AHT	16	8	11	11	10	7	10	13	13	12	11	14	21	22	30	49	51
	(SD)	(2.53)	(2.86)	(4.16)	(4.34)	(4.08	(3.09)	(2.5)	(5.96)	(7.46)	(6.99)	(5.12)	(8.01)	(8.36)	(9.76)	(15.31)	(19.17)	(12.1)
	Min.	10	2.5	2.5	5	0	0	5	7.5	0	0	0	2.5	7.5	7.5	7.5	10	20
	Max.	20	12.5	17.5	20	17.5	10	12.5	27.5	27.5	22.5	20	27.5	35	35	55	75	62.5

AHT, average hearing thresholds (dB); SD, standard deviation.

**Table 4 ijerph-18-06463-t004:** Comparison of average hearing thresholds of respondents unexposed and exposed to noise, depending on the length of work-related exposure to noise (in terms of *p*-values).

Years in Noise at Work		Frequency (kHz)																
		0.125	0.25	0.5	0.75	1	1.5	2	3	4	6	8	9	10	11.25	12.5	14	16
<5 years	Sound pressure level (dB)	NS	NS	0.048	0.003	NS (0.066)	0.021	0.002	NS	NS	0.008	0.004	0.001	0.014	0.001	0.002	NS	NS
5–10 years		NS	NS	NS	0.011	NS	NS	NS	0.026	NS	0.007	0.049	0.003	NS (0.080)	0.037	0.005	NS	NS
<10 years		NS	NS	NS	0.039	0.029	0.013	0.001	0.030	NS	NS (0.069)	NS	0.016	0.022	0.001	0.003	NS	NS

NS: not significant—result of Mann–Whitney test was >0.05. The level of statistical significance was set at 5%.

## Data Availability

The datasets used and analyzed during the current study are available from the corresponding author on reasonable request.

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
