# Peer review of "High-Frequency Audiometry in Women with and without Exposure to Workplace Noise"

_ijerph, 2021, doi:10.3390/ijerph18126463_

Round 1

Reviewer 1 Report

The paper is good and actually in a really good status. I've nothing to add except, maybe, a request of adding a couple lines with a description of workers activities.

Author Response

Dear reviewer,

Thank You for comment.

The review has been edited by IJERPH english editor. We have also added a new text or changes, it‘s highlighted in yellow.

Kind regards

Michaela Škerková

Reviewer 2 Report

The submitted paper is really good and well written. It does not represent a novelty in the sector, but it provides a useful update of the current scientific knowledge, thus expanding its validity.

Author Response

(The authors gave the same response as above.)

Reviewer 3 Report

Many improvements to the manuscript were made. Currently, the manuscript should be corrected according to the comments:

Comment on Response 1

Some of the expected information is included on lines 124-131 (PDF). However, the reference to the EN ISO: 1999 standard is not correct. This standard specifies a method for calculating the expected noise-induced permanent threshold shift in the hearing threshold levels due to various levels and durations of noise exposure and it is not strictly related to the requirement to use hearing protectors. Rather, a short comment would be expected here linking the scope of work of the authors of the manuscript with the method of predicting hearing thresholds given in the standard.

Comment on Response 2

The "dBA" notation was corrected (lines 90-91). In the revised manuscript, however, there are units written as "dB HL". It is also advisable to avoid this form of writing in a research paper. Although you can find examples of the use of "dB HL" as a unit in the published literature standards for audiometric measurements define the hearing level (HL), which should be expressed in "dB". The name of the parameter indicates that the reference is not 20 µPa but it is a hearing threshold. The "dB HL" notation is not needed. Please correct other similar instances of “dB HL”. Please consider whether it is possible to avoid writing "dB HL" in the manuscript.

Comment on Response 6

"Average sound intensity [Pa]" is currently in Table 2. "Sound intensity" does not match "Pa", however. Please do not follow my earlier suggestion that "Pa" should be here, because it is not clear what was to be presented at this point. It is necessary to correct and explain what the authors show here.

Author Response

Thank you for the comments.

It was corrected according to points:

The review has been edited by IJERPH english editor. We have also added a new text or changes, it‘s highlighted in yellow.

Comment on Response 1

Some of the expected information is included on lines 124-131 (PDF).

  • Thank you for this comment.

However, the reference to the EN ISO: 1999 standard is not correct. This standard specifies a method for calculating the expected noise-induced permanent threshold shift in the hearing threshold levels due to various levels and durations of noise exposure and it is not strictly related to the requirement to use hearing protectors. Rather, a short comment would be expected here linking the scope of work of the authors of the manuscript with the method of predicting hearing thresholds given in the standard.

  • Was specified the protection of workers through noise into the methods using hearing protectors Uncorded Polyurethane Foam Earplugs NRR 33dB (lines 141-146 Word, lines 125-130 PDF).
  • The mentioned standard EN ISO 1999: 2013 was removed from Matherials and Methods. It was added to the discussion and it was stated that the aim of our work was not to evaluate possible hearing loss from noise, ie estimating hearing loss due to noise, but to evaluate current hearing loss from noise in the population exposed to work noise and unexposed population living in the same conditions at high frequencies (lines 406-432 Word, lines 354-380 PDF).
  • According to the WHO. Environmental Noise Guidelines for the European Region. Copenhagen: WHO Regional Office for Europe. 2018: Demonstrating a causal link between leisure noise and hearing impairment is problematic due to technical difficulties in accurately quantifying exposure. Hearing loss is usually the result of combined exposure to noise from different sources, ie from the work and the environment and from leisure activities. Noise exposure accumulates throughout life. Therefore, an exposed and unexposed population living in the same region and having the same habits was selected for the work (line 157 Word, line 140-141 PDF).

Comment on Response 2

The "dBA" notation was corrected (lines 90-91). In the revised manuscript, however, there are units written as "dB HL". It is also advisable to avoid this form of writing in a research paper. Although you can find examples of the use of "dB HL" as a unit in the published literature standards for audiometric measurements define the hearing level (HL), which should be expressed in "dB". The name of the parameter indicates that the reference is not 20 µPa but it is a hearing threshold. The "dB HL" notation is not needed. Please correct other similar instances of “dB HL”. Please consider whether it is possible to avoid writing "dB HL" in the manuscript.

  • Thank you for this comments.
  • dB HL was corrected throughout the manuscript as dB.

Comment on Response 6

"Average sound intensity [Pa]" is currently in Table 2. "Sound intensity" does not match "Pa", however. Please do not follow my earlier suggestion that "Pa" should be here, because it is not clear what was to be presented at this point. It is necessary to correct and explain what the authors show here.

  • Thank you for this comments.
  • Sound intensity was corrected for sound pressure.
  • It was compared the difference in the average hearing threshold measured in dB (between the exposed and unexposed groups) The same comparison of average hearing thresholds was performed in the originally used unit Pa (sound pressure). It was sought the difference (whether there were statistically significant differences in dB at the same frequencies as in the comparison in Pa).

Reviewer 4 Report

The manuscript describes measurement of pure-tone hearing thresholds including high-frequencies up to 16 kHz in a sample of 243 women of whom 88 had some noise exposure at work. 

I'm sorry to be negative, but to me, the manuscript is much too long and involved for such a simple study. No firm conclusions can be drawn about the relative impact of noise and age in the samples since the noise-exposure-history of the two groups is not well explained, the exposed group is quite small, and so the conclusions don't warrant the great length of the Results and Discussion.

I encourage the authors to repackage the work as a much shorter, 1-2 page, communication without most of the writing, but with much more clarity about the participants' history of exposure to noise and ototoxic chemicals and use of HPE/PPE. In it, include Figure 1, but the error bars seem odd and are not explained so please make sure they are right and explain what they represent. If proper error bars are included, you can avoid all the p-values (see Cumming, G., Fidler, F., & Vaux, D. L. (2007). Error bars in experimental biology. Journal of Cell Biology, 177(1), 7-11.).

Author Response

Thank you for the comments.

The review has been edited by IJERPH english editor. We have also added a new text or changes, it‘s highlighted in yellow.

It was corrected according to points:

I'm sorry to be negative, but to me, the manuscript is much too long and involved for such a simple study. No firm conclusions can be drawn about the relative impact of noise and age in the samples since the noise-exposure-history of the two groups is not well explained, the exposed group is quite small, and so the conclusions don't warrant the great length of the Results and Discussion.

I encourage the authors to repackage the work as a much shorter, 1-2 page, communication without most of the writing, but with much more clarity about the participants' history of exposure to noise and ototoxic chemicals and use of HPE/PPE. In it, include Figure 1, but the error bars seem odd and are not explained so please make sure they are right and explain what they represent. If proper error bars are included, you can avoid all the p-values (see Cumming, G., Fidler, F., & Vaux, D. L. (2007). Error bars in experimental biology. Journal of Cell Biology, 177(1), 7-11.).

  • The work was shortened by 1 page.
  • Added information in Matherials and Methods: Occupational hygiene inspections are carried out regularly. The only risk factor in workplace was noise (lines 134-135 Word, lines 122-124 PDF).
  • Added information about Hearing protectors were used at the workplace - Uncorded Polyurethane Foam Earplugs NRR 33dB (lines 141-146 Word, lines 125-130 PDF).
  • According to the WHO. Environmental Noise Guidelines for the European Region. Copenhagen: WHO Regional Office for Europe. 2018: Demonstrating a causal link between leisure noise and hearing impairment is problematic due to technical difficulties in accurately quantifying exposure. Hearing loss is usually the result of combined exposure to noise from different sources, ie from the work and the environment and from leisure activities. Noise exposure accumulates throughout life. Therefore, an exposed and unexposed population living in the same region and having the same habits was selected for the work (line 157 Word, line 140-141 PDF).
  • Error bars in graphs were deleted (lines 312 and 353 Word, lines 269 and 308 PDF).
  • Error bars were added to the graphs to better understand the results. These error bars were some lines of standard error. The original tables (Table 1 and Table 3) contained only average values. During the revisions of the article, the tables were supplemented by minimum, maximum and standard deviation. With the use of statistical indicators in the tables, it is no longer necessary to use errors bars and therefore they was removed for better clarity in the graphs.

Round 2

Reviewer 4 Report

The manuscript describes the pure-tone hearing thresholds up to 16 kHz in a sample of 243 women of whom 88 had some noise exposure at work. It has been shortened somewhat from the previous version and is easier to read, but I ask that the authors attempt to reduce the size further.

I couldn’t see whether the participants had potentially been exposed to noise on the day that audiometric testing was conducted. If so, there may be a combination of TTS and PTS in the data. Please can the authors clarify this.

The extent of noise exposure is unclear. In lines 124-140, the authors state that HPE providing up to 33 dB of attenuation was used, alongside noise exposure levels ranging from 87-90 dBA. If both these statements are true, then the workers would have been exposed only to around 60 dBA, which would not cause hearing loss. Given that some differences in hearing threshold were observed between the exposed and non-exposed groups (after grouping according to age), the implication is that HPE was not worn in an effective manner. Please can the authors comment on this and revise the description given to acknowledge the possibility that HPE were not worn reliably.

Table 2 shows the p-values associated with the data presented in Figure 1 and Table 1. The first part of the table presents the significance of the differences in hearing thresholds which are currently presented as “dB”. I presume this will be dB HL, so please can the authors clarify this. The table also presents analyses conducted using hearing thresholds expressed in Pa rather than dB. It isn’t surprising that these are generally less likely to be significant because the conversion to a linear scale will increase the variance in the data, so I ask that the authors remove the lower half of Table 2.

Figure 2 and the associated analyses presented in Tables 3 and 4 do not add meaningful information. I can understand that the authors wished to present findings relevant to the number of years of exposure, however the analyses presented do not allow valid comparisons to be made. The three groups divided according to years of noise at work include women of a range of different ages that may or may not be proportionate to the age range in the non-exposed group. This means that comparisons are not meaningful. Furthermore, since the longer-term noise exposed workers will tend to be older, due to the passage of time, any apparent effects are doubly compromised. Thirdly, the numbers in the three exposure groups are quite different (37, 28 and 13). It is therefore unclear whether differences and lack of differences in comparison to the non-exposed group are a result of these different numbers (and thus differing error terms) or the differences in the means. On the basis of these concerns I ask the authors to remove Figure 2 and Tables 3 and 4 (and related parts of the Discussion). These problems are often insurmountable when there is a combination of chronic exposures and age-related decline. One possible way around them could be to remove participants in the older age group from all four groups in these analyses, since you could then argue that the most likely cases of age-related hearing loss would not be present. Even then, we have a problem in that the higher frequencies (>8kHz), may still be impacted by age-related losses, and the issue of statistical power with different group sizes remains.

A large block of text (Lines 353-379) has been added to the Discussion, but it isn’t relevant to the findings since these were in people wearing HPE. I don’t understand why it was added, and prefer that it be removed again.
